biophysics

microtubules, actin, stochastic processes, biomolecules

**Author for correspondence:**
Marco Saltini
e-mail: marco.saltini@ebc.uu.se

†Current affiliation: Uppsala University, Department of Ecology and Genetics, Animal Ecology, Norbyvägen 18D, 752 36 Uppsala, Sweden.

# Microtubule-based actin transport and localization in a spherical cell

Marco Saltini† and Bela M. Mulder

AMOLF, Science Park 104, 1098 XG Amsterdam, The Netherlands

MS, 0000-0002-5425-9101; BMM, 0000-0002-8620-5749

The interaction between actin filaments and microtubules is crucial for many eukaryotic cellular processes, such as, among others, cell polarization, cell motility and cellular wound healing. The importance of this interaction has long been recognized, yet very little is understood about both the underlying mechanisms and the consequences for the spatial (re)organization of the cellular cytoskeleton. At the same time, understanding the causes and the consequences of the interaction between different biomolecular components are key questions for *in vitro* research involving reconstituted biomolecular systems, especially in the light of current interest in creating minimal synthetic cells. In this light, recent *in vitro* experiments have shown that the actin-microtubule interaction mediated by the cytolinker TipAct, which binds to actin lattice and microtubule tips, causes the directed transport of actin filaments. We develop an analytical theory of dynamically unstable microtubules, nucleated from the centre of a spherical cell, in interaction with actin filaments. We show that, depending on the balance between the diffusion of unbound actin filaments and propensity to bind microtubules, actin is either concentrated in the centre of the cell, where the density of microtubules is highest, or becomes localized to the cell cortex.

## 1. Introduction

Microtubules and actin filaments are dynamic polymers and components of the eukaryotic cytoskeleton. Both are involved in spatial processes at the scale of the cell. Microtubules are best known for their role in motor protein-mediated directed intracellular transport and forming the mitotic spindle—the machinery for segregating the duplicated chromosomes prior to cell division, while actin is strongly associated with cell locomotion and deformation, e.g. during many developmental processes. Although historically, actin and microtubules have been studied independently from each other, it has more recently been acknowledged that the interaction between these

two species is crucial in processes such as, among others, cell division, cell growth and migration, cellular wound healing and cell polarization [1–5].

Actin and microtubules can interact in many different ways: examples are steric repulsion, passive crosslinking and microtubule growth guidance by actin bundles [6,7]. The interaction between actin filaments and microtubules has been shown to influence the spatial organization of each other [8]. Here, we focus specifically on the interactions mediated by a class of proteins collectively called cytolinkers. Some cytolinkers, like, e.g. MACF1, have been observed to bind both to actin filaments and microtubules [9–12], suggesting that the spatial organization of the former can be influenced by the dynamics of the latter.

This possibility was explored in recent *in vitro* experiments [10,13,14] employing the engineered protein construct TipAct [6] specifically designed to bind to both actin and microtubules. These experiments revealed a dual action of TipAct. It was shown to bind to the plus-end of growing microtubules mediated by the presence of end binding protein EB3. This results in a decrease of the microtubule growth speed, and an increase in their catastrophe rate, both effects effectively shortening the microtubule. On the other hand, TipAct also tracks the growing microtubule tip and, hence, transports any bound actin in the direction of microtubule growth. Experiments in planar quasi-two-dimensional geometries as well as spherical droplets revealed a distinct TipAct-dependent impact of microtubules on the spatial distribution of the actin.

While experimental and theoretical studies aimed at understanding the underlying microscopic mechanism behind the transport of actin through TipAct-mediated coupling to dynamic microtubules are currently underway [15], the question of the macroscopic effects of the resulting actin transport have to date not been addressed. Here, we consider the latter question in a minimal model consisting of dynamic microtubules nucleated from a point-like microtubule organizing centre (MTOC) [16] located at the centre of a spherical cell. This cell contains a finite amount of actin which either diffuses freely in the cytosol or directly binds to microtubule tips, i.e. the presence of the cytolinker mediating this coupling is implicit. Our primary aim is to elucidate the resulting spatial distribution of the actin and to determine which of the model parameters are the main determinants of this distribution.

This paper is structured as follows. We first introduce a stochastic model of microtubules undergoing dynamic instability in a three-dimensional cell. Within this cell actin filaments can either diffuse free of any driving forces or be bound to microtubule plus ends and subsequently transported towards the boundary of the cell. We show that, under the assumption that the dynamics of microtubules is not influenced by the binding of the actin filaments, the dynamic equations of the model decouple. Under this assumption, we obtain numerical solutions for the spatial distribution of both the microtubules and the actin. We show, supported by a dimensional analysis, that while the spherical geometry by default would promote the actin to be concentrated in the cell centre where the density of microtubules is highest, a combination of low actin diffusivity coupled to a high propensity to bind microtubules causes the actin distribution to be cortical. Finally, we discuss why, for model parameters in the range of biological relevance, the model cannot naively be approximated by an advective-diffusive model in which the microtubule dynamics acts as an overall force field that pushes the actin filaments towards the surface of the cell.

# 2. Methods

In this section, we set up a stochastic model of dynamic microtubules undergoing dynamic instability in a spherical cell, where actin filaments diffuse and can interact with the plus end of microtubules. Because preliminary experiments have revealed that the interaction between actin and microtubules via TipAct can change the dynamic properties of the latter [13], we first define a general model that accounts for this change in the dynamics. Nevertheless, our main interest lies in the actin transport mechanism observed in the experiments performed in the droplet, where no change in the dynamics of microtubules has been recorded [14]. Therefore, we will provide a solution of the model under the assumption that the dynamic properties of microtubules do not change as a consequence of the interaction with actin filaments. Although we present a three-dimensional model, some of the results presented in this work can easily be generalized to fewer dimensions.

## 2.1. The model: dynamic microtubules and diffusing actin

The model is based on the Dogterom and Leibler model for microtubules undergoing dynamic instability [17]. It consists of $M$ microtubules undergoing dynamic instability in an homogeneous three-dimensional

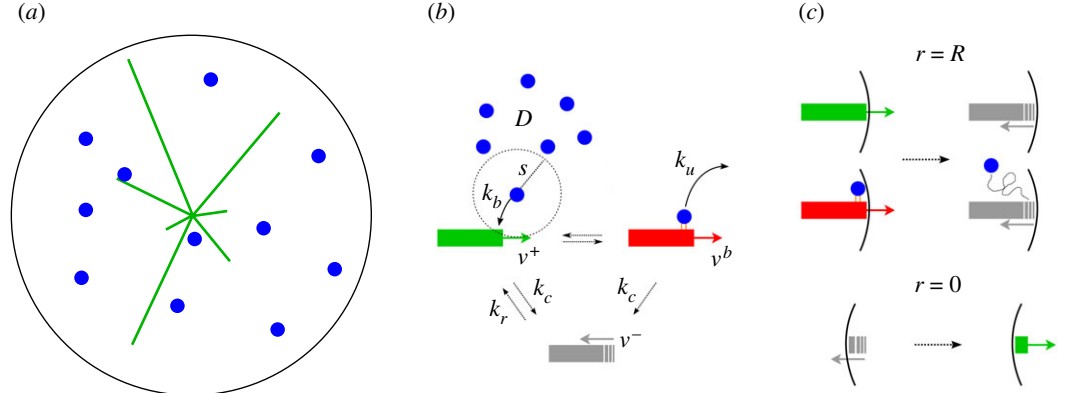

**Figure 1.** Schematic of the model. (*a*) Microtubules (green lines) undergoing dynamic instability and actin filaments (blue dots) diffusing in a three-dimensional sphere. (*b*) Schematic of the dynamics of microtubules and their interaction with actin filaments. (*c*) Boundary conditions at the cortex of the cell and at the centre.

sphere of radius $R$, interacting with $A$ actin filaments diffusing with diffusion constant $D$, and with reflecting boundary conditions at the boundary $r = |\mathbf{r}| = R$. As we are interested in studying the interaction between actin and microtubules, and in particular in the transport of actin by the plus end of microtubules, in our model we ignore both the polymerization/depolymerization and the nucleation of the actin filaments, and we model them as dimensionless particles, see figure 1*a*.

All microtubules are isotropically nucleated at position $\mathbf{r} = 0$ in the growing state, with growing velocity $\mathbf{v}^+ = v^+ \hat{\mathbf{r}}$, where $\hat{\mathbf{r}}$ is the unit vector in the radial direction. Microtubules can switch from the growing to the shrinking state with constant catastrophe rate $k_c$. Then, microtubules shrink with velocity $\mathbf{v}^- = -v^- \hat{\mathbf{r}}$. A shrinking microtubule either switches from shrinking to growing state with constant rescue rate $k_r$, or it completely depolymerizes. When complete depolymerization occurs, reflective boundary conditions at $r = 0$ for microtubules implement their immediate re-nucleation in the growing state. Finally, we impose reflective boundary conditions at $r = R$ as well, with both growing and bound microtubules switching to the shrinking state directly upon reaching the cortex of the cell, see figure 1*b,c*.

When the plus-end of a microtubule is within a range of interaction $s$ of an actin filament, the microtubule tip and the filament can interact, leading the actin filament to bind with binding rate $k_b$, and the microtubule entering the bound state. The bound filament is subsequently transported by the microtubule plus end towards the cortex of the cell, with transport velocity $\mathbf{v^b} = v^b \hat{\mathbf{r}}$, where $v^b < v^+$, in accordance with experimental measurements [13]. These experiments also revealed that the interaction between actin and microtubules has a second effect on microtubule dynamics, i.e. it increases the catastrophe rate. Here, however, for simplicity's sake we decide to ignore this effect. In fact, in the next section we will show how this model always reaches the steady state and, as a consequence, the properties of microtubules are solely defined by the *ratio* between growing speed and catastrophe rate. We, therefore, decided to keep the catastrophe rate constant, and change the growing speed in order to take into account the changed dynamics observed in the experiments. A bound actin filament can unbind in three distinct ways: (i) it simply detaches with a constant unbinding rate $k_u$, (ii) the microtubule to which it is bound undergoes a catastrophe with rate $k_c$ and releases the filament, (iii) or when the microtubule hits the surface of the cell, see figure 1*b,c*. The values of the dynamic parameters used in the model were chosen in agreement with the experimental measurements, see table 1.

## 2.2. Dynamic equations

We assume that the interaction process which leads to actin binding is fast compared to both diffusion of the actin and the growth of a microtubule. Moreover, we assume that the interaction range $s$ is small enough that the density distribution of growing microtubule plus ends is approximately constant within it. Therefore, if $m^+(t, \mathbf{r})$ is the distribution of the position of free growing microtubule tips, and $a(t, \mathbf{r})$ the distribution of the position of free actin filaments, the overall interaction strength can be given by

$$K_{\text{int}}(t, \mathbf{r}) = k_b s\, m^+(t, \mathbf{r}) a(t, \mathbf{r}). \tag{2.1}$$

**Table 1.** Model parameters. (The choice for the numerical values is in agreement with the experimental measurements of the same quantities [13].)

| parameter | description | numerical value | units |
| --- | --- | --- | --- |
| $v^+$ | free-growth speed | 0.05 | $\mu m\,s^{-1}$ |
| $v^b$ | transport speed | 0.03 | $\mu m\,s^{-1}$ |
| $k_c$ | catastrophe rate | 0.005 | $s^{-1}$ |
| $k_b$ | binding rate | 0.8 | $s^{-1}$ |
| $s$ | actin-microtubule interaction volume | 0.002 | $\mu m^3$ |
| $k_u$ | unbinding rate | 0.009 | $s^{-1}$ |
| $D$ | free actin diffusion coefficient | 0.5 | $\mu m^2\,s^{-1}$ |
| $R$ | radius of the cell | 10 | $\mu m$ |
| $M$ | total number of microtubules | $10^4$ | — |
| $A$ | total number of actin filaments | $5 \times 10^4$ | — |

Then, the dynamic equations for actin filaments and microtubules—including shrinking $m^-(t, \mathbf{r})$ and bound $b(t, \mathbf{r})$ microtubules as well, are:

$$4\pi r^2 \frac{\partial m^+(t, \mathbf{r})}{\partial t} = -\mathbf{v}^+ \cdot \nabla 4\pi r^2 m^+(t, \mathbf{r}) - k_c 4\pi r^2 m^+(t, \mathbf{r})$$
$$- k_b s\, 4\pi r^2 m^+(t, \mathbf{r})a(t, \mathbf{r}) + k_u 4\pi r^2 b(t, \mathbf{r}) + k_r 4\pi r^2 m^-(t, \mathbf{r}), \quad (2.2)$$

$$4\pi r^2 \frac{\partial m^-(t, \mathbf{r})}{\partial t} = -\mathbf{v}^- \cdot \nabla 4\pi r^2 m^-(t, \mathbf{r})$$
$$+ k_c 4\pi r^2 [m^+(t, \mathbf{r}) + b(t, \mathbf{r})] - k_r 4\pi r^2 m^-(t, \mathbf{r}), \quad (2.3)$$

$$4\pi r^2 \frac{\partial b(t, \mathbf{r})}{\partial t} = -\mathbf{v}^\mathbf{b} \cdot \nabla 4\pi r^2 b(t, \mathbf{r}) - (k_c + k_u) 4\pi r^2\, b(t, \mathbf{r})$$
$$+ k_b s\, 4\pi r^2 m^+(t, \mathbf{r})a(t, \mathbf{r}), \quad (2.4)$$

$$\frac{\partial a(t, \mathbf{r})}{\partial t} = D\nabla^2 a(t, \mathbf{r}) - k_b s\, m^+(t, \mathbf{r})a(t, \mathbf{r}) + (k_u + k_c)b(t, \mathbf{r}). \quad (2.5)$$

The first three equations are the evolution transport equations for the radial distributions of the plus end of growing microtubules, shrinking microtubules and bound microtubules, respectively. The last equation is a diffusion equation for the free actin particles with a loss term due to the capture of actin filaments by microtubule tips, and a source term due to the release of actin from the microtubule tip to the pool.

Because the system possesses spherical symmetry, we make use of spherical coordinates, i.e. $\mathbf{r} = (r, \theta, \phi)$. Note that, given our assumptions of homogeneity and isotropy, all quantities of the model only depend on the radial coordinate $r$. Furthermore, it has been shown that the length distribution of microtubules undergoing dynamic instability in a confined volume always reaches a steady state, regardless of the choice of the dynamic parameters [18]. More generally, the length distribution of microtubules undergoing dynamic instability in the presence of any limiting factor such as, e.g. finiteness of free tubulin, always reaches a steady state [19]. Hence, we reasonably assume that our system always reaches the steady state, and we restrict the study of equations (2.2)–(2.3) to that situation. Finally, because in the experiments no rescues have been observed [13], we set $k_r = 0$. In this way, once a microtubule undergoes a catastrophe, its fate is determined as it cannot be rescued. Therefore, we make the final assumption that $v^- \gg v^+$, i.e. as soon as a microtubule undergoes a catastrophe, it suddenly completely depolymerizes, and then, it is immediately re-nucleated again. In this way, the number of microtubules in the shrinking state—and hence the related distribution, vanishes and equation (2.3) is redundant. This choice is also motivated by the fact that backward transport of actin filaments by shrinking has been very rarely observed. Therefore, including shrinking microtubules would only raise the complexity of the model without providing further insight.

Under these assumptions, we can rewrite equations (2.2)–(2.3) in the steady state as:

$$0 = -v^+ \frac{\mathrm{d}}{\mathrm{d}r}[r^2 m(r)] - k_c r^2 m(r) - k_b s\, r^2 m(r)a(r) + k_u r^2 b(r), \tag{2.6}$$

$$0 = -v^b \frac{\mathrm{d}}{\mathrm{d}r}[r^2 b(r)] - (k_c + k_u)r^2 b(r) + k_b s\, r^2 m(r)a(r), \tag{2.7}$$

$$0 = D\frac{1}{r^2}\frac{\mathrm{d}}{\mathrm{d}r}\left[r^2 \frac{\mathrm{d}}{\mathrm{d}r}a(r)\right] - k_b s\, m(r)a(r) + (k_u + k_c)b(r), \tag{2.8}$$

where $m(r) \equiv m^+(r)$. This set of ordinary differential equations is supplemented by the boundary conditions defined by the properties of the model. Indeed, the sudden renucleation of every microtubule that undergoes a catastrophe implies

$$v^+ m(0) = v^+ m(R) + v^b b(R) + k_c M \tag{2.9}$$

and

$$b(0) = 0. \tag{2.10}$$

The release of actin filaments from bound microtubules when they reach the cell surface, together with the reflective boundary condition for the free diffusing actin, imply

$$D\nabla a(r) \cdot \hat{\mathbf{r}}|_{r=R} = v^b b(R). \tag{2.11}$$

Finally, conservation of probability implies a normalization condition for both actin filaments and microtubules:

$$4\pi \int_0^R \mathrm{d}r\, r^2[m(r) + b(r)] = M \tag{2.12}$$

and

$$4\pi \int_0^R \mathrm{d}r\, r^2[a(r) + b(r)] = A. \tag{2.13}$$

## 2.3. Solution

Multiplying equation (2.6) by $r^2$ and adding it to equation (2.5) yields

$$D\frac{\mathrm{d}}{\mathrm{d}r}\left[r^2 \frac{\mathrm{d}}{\mathrm{d}r}a(r)\right] = v^b \frac{\mathrm{d}}{\mathrm{d}r}[r^2 b(r)], \tag{2.14}$$

which implies

$$r^2\left[\frac{\mathrm{d}}{\mathrm{d}r}a(r) - \frac{v^b}{D}b(r)\right] = \mathrm{const.}, \tag{2.15}$$

and, using the boundary condition (2.9),

$$\frac{\mathrm{d}}{\mathrm{d}r}a(r) = \frac{v^b}{D}b(r) \qquad \forall r \in [0, R]. \tag{2.16}$$

Because $b(r)$ is a distribution and therefore, always positive, from equation (2.14) it follows that $a(r)$ is monotonically increasing in the radial direction. Similarly, if we add equation (2.4) to equation (2.5), we find

$$\frac{\mathrm{d}}{\mathrm{d}r}r^2[v^+ m(r) + v^b b(r)] = -k_c r^2[m(r) + b(r)]. \tag{2.17}$$

Given that we are interested in the transport mechanism rather than in the change of microtubule dynamics, we will hereafter work under the assumption that the dynamic properties of the microtubules do not change upon binding actin filaments. This means that we assume

$$v^+ = v^b \equiv v. \tag{2.18}$$

In this regime, equation (2.15) is solvable for $m(r) + b\,(r)$, and the solution is

$$m(r) + b(r) = \frac{m_0}{4\pi r^2}\, \mathrm{e}^{-(k_c/v)r}, \tag{2.19}$$

where we used the normalization condition (2.10), i.e. $M = 4\pi \int_0^R \mathrm{d}r\, r^2 [m(r) + b(r)]$, to find the integration constant:

$$m_0 = \frac{k_c}{v}(1 - \mathrm{e}^{-(k_c/v)R})^{-1}M. \tag{2.20}$$

As equation (2.17) is the steady-state length distribution for non-interacting microtubules [17], we observe that if the interaction between actin filaments and microtubules does not affect the dynamic properties of the latter, the microtubule steady-state length distribution remains the same as in the non-interacting system. Interestingly, this result is not dependent on the kind of process behind the switching from the growing to the bound state and vice versa, as it only depends on the growing speed and the catastrophe rate of microtubules.

Therefore, equations (2.14) and (2.17) allow us to disentangle the set of differential equations (2.4)–(2.6), and to write a closed-form expression for $a\,(r)$:

$$a''(r) + \left[\frac{2}{r} + \frac{k_u + k_c}{v} + \frac{k_b s}{v}a(r)\right]a'(r) - \frac{k_b s}{D}\frac{m_0}{4\pi r^2}\,\mathrm{e}^{-(k_c/v)r}a(r) = 0. \tag{2.21}$$

Even though equation (2.19) is still not analytically solvable, its non-dimensionalization highlights relevant features. Indeed, we first set the unit of length for microtubules:

$$\bar{l} = \frac{v}{k_c}. \tag{2.22}$$

Then, we define the dimensionless parameters: transport length $\tau = k_c/(k_u + k_c)$, binding rate $\beta = k_c^2 k_b s / v^3$ and diffusion coefficient $\delta = k_c\, D/v^2$. We also introduce the dimensionless length $x = r/\bar{l}$, and its distribution $f(x) = \bar{l}^3 a(r)$. The equation for $f$ becomes

$$f''(x) + \left[\frac{2}{x} + \frac{1}{\tau} + \beta f(x)\right]f'(x) - \frac{\beta}{\delta}\frac{\mu_0}{4\pi x^2}\,\mathrm{e}^{-x}f(x) = 0, \tag{2.23}$$

where $\mu_0 = \bar{l}m_0$. Equation (2.21) shows that once the typical length for microtubules and the radius of the cell are set, the behaviour of the distribution of the actin filaments is uniquely determined by three parameters, namely the transport length $\tau$, the binding rate $\beta$ and the diffusion coefficient $\delta$.

We can make equation (2.21) suitable for numerical integration by working with the cumulative function

$$F(x) = \int_0^x \mathrm{d}y\, 4\pi y^2 f(y), \tag{2.24}$$

from which it follows:

$$f(x) = \frac{1}{4\pi x^2}F'(x). \tag{2.25}$$

Indeed, equation (2.21) becomes

$$F'''(x) + \left[-\frac{2}{x} + \frac{1}{\tau} + \beta\frac{F'(x)}{4\pi x^2}\right]F''(x) + \left[\frac{2}{x^2} - \frac{2}{\tau x} - \frac{2\beta}{x}\frac{F'(x)}{4\pi x^2} - \frac{\beta}{\delta}\frac{\mu_0}{4\pi x^2}\,\mathrm{e}^{-x}\right]F'(x) = 0, \tag{2.26}$$

with boundary conditions

$$\left[F''(x) - \frac{2}{x}F'(x)\right]_{x=0} = 0, \tag{2.27}$$

and

$$F(R/\bar{l}) = q, \tag{2.28}$$

coming from equations (2.8) and (2.11), respectively, and where $q$ is the number of free actin filaments. Note that, although $q$ is an unknown quantity, we can numerically find its value by observing that

$$q = A - \int_0^{R/\bar{l}} \mathrm{d}x\, 4\pi x^2 \delta f'(x). \tag{2.29}$$

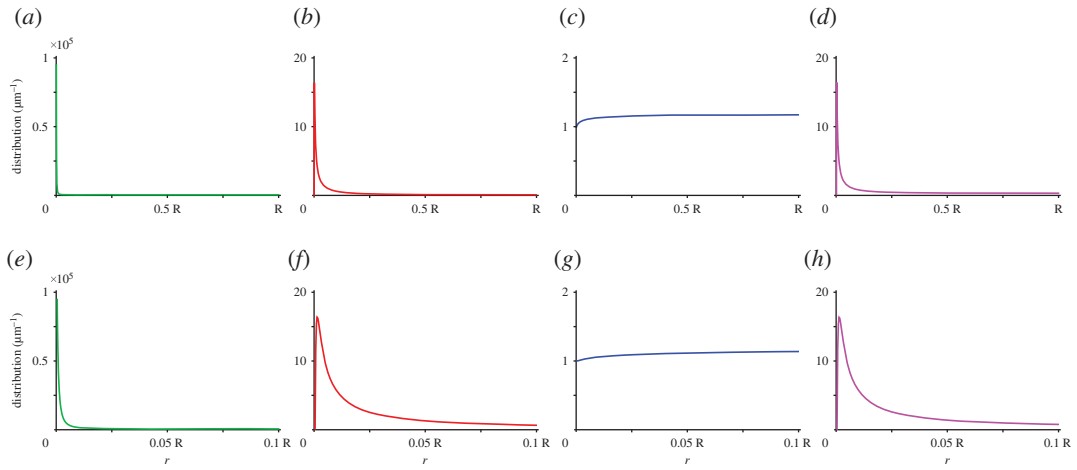

**Figure 2.** Distribution of the positions of (*a*) growing microtubule plus ends, (*b*) bound microtubule plus ends, (*c*) free actin filaments and (*d*) all actin filaments (free and bound). (*e*–*h*) The above distributions in the centre of the cell, when $r \in (0, 0.1\ R)$.

As equation (2.24) is a third-order differential equation, a third boundary condition is in principle required for its solution. However, any integration constant provided by the third condition would be cancelled out by subsequently taking the derivative to find $f(x)$.

## 3. Results

Here, we show the results of our model under the assumption that microtubules do not change their dynamic properties when they are bound to actin filaments, i.e. with the parameter choice in equation (2.16). The were obtained through numerical integration of equation (2.24), with dynamic parameters listed in table 1. In particular, our choice for the growing speed is $v^+ = v^b = v = 0.05\ \mu\mathrm{m\,s^{-1}}$.

Figure 2 shows the distribution of the positions of the tips of the growing and bound microtubules, the free actin filaments, and all actin filaments (both free and bound), respectively. The figure shows that in the biologically relevant range of parameters, the interaction between actin and microtubules plays a minimal role in changing the steady-state distribution of the free actin filaments, as the resulting distribution is roughly uniform (figure 2*c,g*). However, if we consider the total distribution of actin filaments—i.e. including both free and bound filaments, the resulting distribution exhibits a marked peak at the centre of the cell (figure 2*d,h*). The peak is caused by the very high density of microtubule tips at $r \to 0$ owing to the divergence of the density of growing microtubules at the origin. This high density implies an overall high binding rate for the free actin filaments that, therefore, are trapped at the centre of the cell and cannot diffuse away from it. The roughly uniform distribution for the free actin filaments, instead, is a consequence of the high value of the diffusion constant of the system. Indeed, under these conditions any transported filaments, can very quickly redistribute all over the cell volume, without exhibiting a clear localization at the cortex as we could expect in a transport mechanism of this kind. Finally, owing to the dilution of microtubule plus ends caused by their radial orientation from the centre of the cell, we observe that the distribution of microtubules—both growing and bound, very quickly decreases to low values, see figure 2*a,b,e,f*. This, however, does not mean that microtubules are short compared to the radius of the cell. Indeed, for our choice of model parameters, the typical length of microtubules actually equals the radius of the cell, i.e. $v/k_c = R$. The observed diluted microtubule density at the cortex is solely a consequence of the three-dimensional geometry.

To test whether the binding rate and the diffusion coefficient play an important role in the distribution of the actin filaments, we numerically solve equation (2.21) for different $\tau$, $\beta$ and $\delta$. As expected, figure 3*a,b* shows that both a high binding rate and a low diffusion constant have the effect of localizing free acting filaments closer to the cell surface, while changing the transport length $\tau$ does not seem to have a significant influence on $f(x)$, see figure 3*c*. Figure 3*a,b* highlights another interesting fact: except for very high values of $\beta$, scaling $\beta$ with a certain factor $z$, has roughly the same effect on the actin distribution as scaling $\delta$ with the factor $1/z$. This means that there exists a range of values for parameter $\beta$ such that

$$\beta f(x) \ll \frac{2}{x} + \frac{1}{\tau}, \tag{3.1}$$

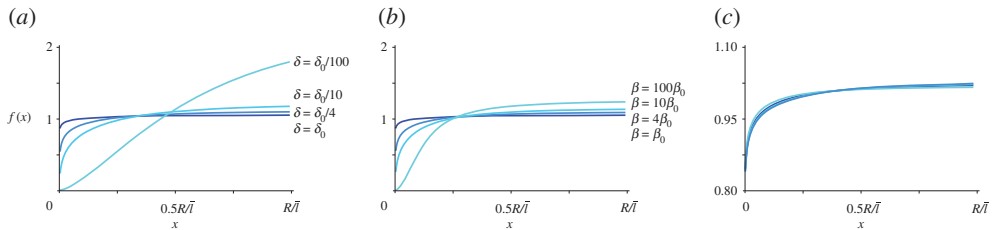

**Figure 3.** Distribution of the positions of free actin filaments when (*a*) $\delta$ is tuned, (*b*) $\beta$ is tuned and (*c*) $\tau$ is tuned ($\tau = 0.2$, $\tau = 0.36$, $\tau = 0.52$, $\tau = 0.68$). Reference values are $\delta_0 = 1$, $\beta_0 = 3.2 \times 10^{-4}$, $\tau_0 = 0.36$, from table 1.

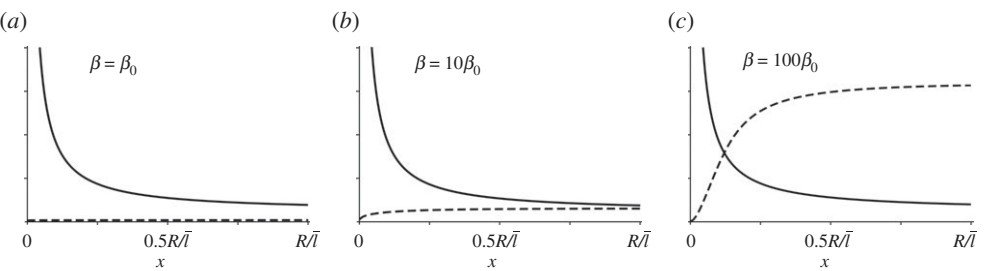

**Figure 4.** Comparison between (solid line) $2/x + (1/\tau)$, and (dashed line) $\beta f(x)$ for three different choices of $\beta$. (*a*) For model parameters in the biologically range of values, the latter is neglectable compared to the former. Reference values are $\delta_0 = 1$, $\beta_0 = 3.2 \cdot 10^{-4}$, $\tau_0 = 0.36$, from table 1.

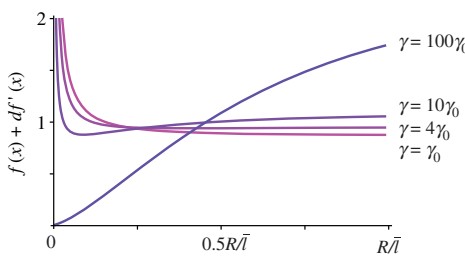

**Figure 5.** Distribution of the cumulative positions of actin filaments, bound and free, for four different choices of $\gamma = \beta/\delta$. The reference value is $\gamma_0 = \beta_0/\delta_0 = 3.2 \times 10^{-4}$, from table 1.

see figure 4. As a consequence, for small $\beta$, equation (2.21) can be approximated with

$$f''(x) + \left(\frac{2}{x} + \frac{1}{\tau}\right) f'(x) - \frac{\beta}{\delta} \frac{\mu_0}{4\pi x^2} e^{-x} f(x) = 0, \tag{3.2}$$

without significant loss of accuracy in the solution. Intriguingly, the biologically relevant limit for $\beta$ is the limit in equation (3.1). Therefore, from equation (3.2) and figure 3*c*, we can observe that the system is fully characterized by only one parameter: the binding/diffusion ratio $\gamma \equiv \beta/\delta$. Figure 5 shows the distribution of all actin filaments—both free and bound to microtubules, for different choices of $\gamma$. In particular, we can observe that a high binding/diffusion ratio increases the actin density at the surface of the cell, while decreasing the effect of the spherical geometry of trapping actin at the centre of the cell. In fact, for $\gamma = 0.032$, free actin filaments can no longer redistribute over the whole volume of the cell after their release from microtubule tips. The consequence is that, although the distribution of growing microtubules remains divergent for $r \to 0$, there are no free actin filaments close to the origin to attach to. Thus, a high binding rate, promoting the transport of the filaments towards the cortex as it enhances the interaction of the latter with the microtubule tips, coupled to a low diffusion coefficient, preventing the actin redistribution over the whole volume of the cell including the central volume, can create an actin density profile with a maximum at the cortex.

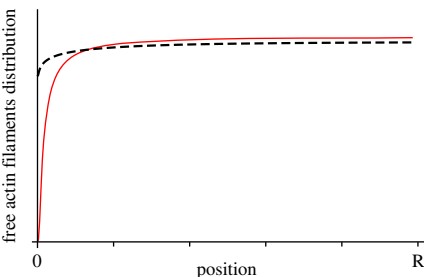

**Figure 6.** Distribution of the position of free actin filaments, for the transport-diffusion model (dashed line), and for the advection–diffusion model (solid line), for model parameters from table 1, and $u = 5 \times 10^{-3}$ μm$^2$ s$^{-1}$.

# 4. Comparison to an advection–diffusion model

At first sight, one could argue that our proposed transport mechanism could well be approximated by a diffusive–advective mechanism with a velocity field directed towards the cortex of the cell with a speed proportional to the density of microtubules. In this case, microtubules act as a position-dependent background that push actin filaments towards the external volume of the cell.

Thus, the steady-state dynamics of the system is described by the equation

$$0 = D\nabla^2 a(\mathbf{r}) - \nabla \cdot [\mathbf{V}(\mathbf{r})a(\mathbf{r})], \tag{4.1}$$

with $\mathbf{V}(\mathbf{r}) = u\,m(\mathbf{r})\hat{\mathbf{r}}$, and where $u$ is an appropriate constant. The density of microtubules, in the absence of binding and unbinding to actin filaments, is the steady-state density

$$m(r) = \frac{m_0}{4\pi r^2}\,\mathrm{e}^{-(k_c/v)r},$$

already discussed in equation (2.17). Conservation of the number of actin filaments imposes the following boundary and normalization conditions for $a\,(\mathbf{r})$:

$$D\nabla a(r) \cdot \hat{\mathbf{r}}|_{r=R} = um(R)a(R) \tag{4.2}$$

and

$$A = \int_0^R \mathrm{d}r\, 4\pi r^2 a(r). \tag{4.3}$$

By combining equations (4.1), (4.2) and (4.3), we find the expression for the density of actin filaments, i.e.

$$a(r) = a_0 \exp\left(\frac{u}{D}\int_0^r \mathrm{d}t\, \frac{m_0}{4\pi t^2}\,\mathrm{e}^{-(k_c/v)t}\right), \tag{4.4}$$

where

$$a_0 = A\left[\int_0^R \mathrm{d}r\, \exp\left(\frac{u}{D}\int_0^r \mathrm{d}t\, \frac{m_0}{4\pi t^2}\,\mathrm{e}^{-(k_c/v)t}\right)\right]^{-1}. \tag{4.5}$$

The integral in equation (4.4) diverges at $r = 0$. Therefore, the density of actin filaments approaches zero as $r \to 0$ in contrast with the observed finite density of free actin filaments of the transport-diffusion model where the presence of actin filaments at the centre of the cell was ensured by the release of trapped filaments by microtubule tips, see figure 6. Intriguingly, this discrepancy is purely a consequence of the three-dimensional geometry of the system. Indeed, in one dimension, it is possible to show that the advective–diffusive model can very well approximate the transport-diffusion mechanism for a suitable choice of $u$, as we show in appendix A.2.

# 5. Discussion

We introduced a minimal model for the interplay between actin filaments and dynamical microtubules, based on the experimentally observed TipAct-mediated interaction between actin filaments and

microtubule plus ends, and the subsequent transport of the former by the latter during microtubule polymerization. Focusing on the question to what extent such a transport mechanism could spatially reorganize the cytoskeleton, we identified the ratio of the binding propensity of actin filaments to the microtubules over their free diffusion coefficient as the key parameter determining the spatial organization of actin filaments. Our analysis showed that a high binding/diffusion ratio can overcome the default trapping of actin filaments at the microtubule-dense centre of the cell, causing actin re-localization to the cell cortex.

At first sight, one might naively argue that this transport mechanism could well be approximated by a diffusion mechanism with a velocity field directed towards the cortex of the cell with an advective speed proportional to the density of microtubules. However, it is readily seen that this system would exhibit a divergent pushing force at $\mathbf{r} \to 0$, as $m(\mathbf{r})$ diverges at the centre of the cell because of the three-dimensional geometry. That would result in a complete absence of free actin filaments at the centre of the cell, and in a monotonic increase of the total actin distribution from the centre to the boundary. This is in contrast with our full transport model, where for low binding/diffusion ratio, we observe trapping of actin at the centre of the cell owing to the high density of microtubules.

However, it is important to underline that the very high peak in the total actin distribution close to $r = 0$ is purely the consequence of divergence of the distribution of free microtubules. This divergence, mathematically inherent to the design of our model, could be in principle be removed by assuming a finite size for the centrosome located at $r = 0$, and treating microtubules as objects with a finite diameter of about 25 nm [20] rather than one-dimensional entities. Further analytical investigations should therefore aim at testing to what extent the details of the geometry influences the result.

Although our model gives us insights about how to tune the binding/diffusion ratio in order to obtain localization of actin at the cortex of the cell, it as yet fails in giving a complete description of the spatial organization of the cytoskeleton that includes the interaction among its components. Indeed, we limited ourselves to microtubules in the bounded-growth regime and in particular in the limit of fast depolymerization. Including shrinking microtubules and rescues from the shrinking to the growing state in the system, would add a further degree of complexity to the set of equations (2.4)–(2.6), making them no longer analytically tractable. Furthermore, in our model, we have ignored both actin–actin interactions, driving the organization in networks at the cortex, as well as the dynamic instability and the nucleation of actin filaments. Therefore, a full description of the system, including shrinking microtubules and interaction between different actin filaments, most likely requires a more brute-force simulation approach.

Nevertheless, our theoretical predictions on the response of the system to the change of the binding/diffusion ratio, could in principle be validated by experiments, as transport in the three-dimensional domain has recently been observed in droplets [14]. While directly manipulating the transport and binding rates may be harder to achieve experimentally, tuning the diffusion coefficient by changing the viscosity of the medium seems feasible. Another possibility could be to change the typical length of actin filaments. Indeed, it has been observed that the diffusion coefficient of an actin filament is inversely proportional to its length [21]. Engineering longer or shorter actin filaments could then test our hypothesis that a high diffusion coefficient is correlated to the localization of actin close to the cortex.

While our model is neither aimed at nor able to describe currently known mechanisms of microtubule-actin organization *in vivo*, we hope that our work could serve as an inspiration for future more integrated analytical, computational and experimental research on how the interactions between different cytoskeletal components determine spatial structure of the cytoskeleton. At the very least, it may provide useful insights towards the design of reconstituted minimal systems aimed at reproducing some traits of living cells, which are dependent on a properly organized actin cortex. In that light it would also be interesting in future to consider, e.g. the role of actin nucleators such as formins, which have been observed to interact with microtubules, and specifically with growing microtubule plus ends [22].

Data accessibility. This article has no additional data.

Authors' contributions. M.S. designed the model and carried out the formal analysis. B.M.M. supervised and coordinated the study. M.S. and B.M.M. conceived the study, wrote the manuscript and edited the manuscript.

Competing interests. We declare we have no competing interests.

Funding. The work of M.S. was supported by the ERC 2013 Synergy Grant MODELCELL. The work of B.M.M. is part of the research programme of the Dutch Research Council (NWO).

Acknowledgements. We acknowledge many helpful discussions with Celine Alkemade, Gijsje Koenderink and Marileen Dogterom (Delft Technical University).

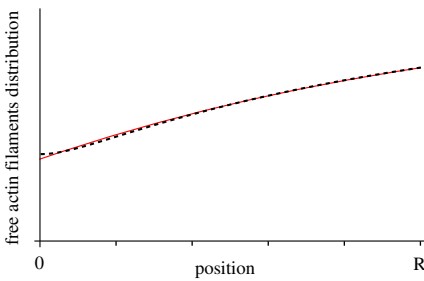

**Figure 7.** Distribution of the position of free actin filaments, for the transport-diffusion model (black dashed line), and for the advection–diffusion model (red solid line), for model parameters from table 1, and $u = 10^{-4}\ \mu m^2\ s^{-1}$.

# Appendix A

In this section, we show that, in one dimension, our model can well be approximated by an advective–diffusive model where microtubule density acts as an overall velocity field that pushes actin filaments towards the cortex of the cell.

## A.1. One-dimensional transport of actin filaments

In the one-dimensional case, the steady-state dynamic equations (2.4)–(2.6), under the assumption that no changes occur to the dynamics of microtubules when they bind actin filaments, can be rewritten as:

$$0 = -v\frac{\mathrm{d}}{\mathrm{d}x}m(x) - k_c m(x) - k_b s\, m(x)a(x) + k_u b(x), \tag{A 1}$$

$$0 = -v\frac{\mathrm{d}}{\mathrm{d}x}b(x) - (k_c + k_u)b(x) + k_b s\, m(x)a(x), \tag{A 2}$$

$$0 = D\frac{\mathrm{d}^2}{\mathrm{d}x^2}a(x) - k_b s\, m(x)a(x) + (k_u + k_c)b(x), \tag{A 3}$$

with boundary and normalization conditions:

$$vm(0) = vm(R) + vb(R) + k_c M, \tag{A 4}$$

$$b(0) = 0, \tag{A 5}$$

$$D\frac{\mathrm{d}}{\mathrm{d}x}a(x)\Big|_{x=R} = vb(R), \tag{A 6}$$

$$\int_0^R \mathrm{d}x\,[m(x) + b(x)] = M, \tag{A 7}$$

$$\int_0^R \mathrm{d}x\,[a(x) + b(x)] = A. \tag{A 8}$$

Similarly to the three-dimensional case, equations (2.4)–(2.6) can be disentangled to find a closed-form expression for the density of free actin filaments, i.e.

$$a''(x) + \left[\frac{k_u + k_c}{v} + \frac{k_b s}{v}a(x)\right]a'(x) - \frac{k_b s}{D}\frac{k_c}{v}M\frac{e^{-(k_c/v)x}}{1 - e^{-(k_c/v)R}}a(x) = 0. \tag{A 9}$$

The latter equation can be numerically solved to obtain the density of actin filaments, see figure 7.

## A.2. One-dimensional advection–diffusion model

We now show that this result can be well approximated by an advective–diffusive mechanism in which microtubules act as a velocity field that pushes actin towards the cortex of the cell. Hence, in this case, we can consider actin undergoing a diffusive process with a drift force proportional to the microtubule density $m(x)$, and with reflective boundary conditions at both boundaries $x = 0, R$.

Therefore, steady-state dynamic equations for microtubules and actin are

$$0 = -v\frac{d}{dx}m(x) - k_c m(x) \tag{A 10}$$

and

$$0 = D\frac{d^2}{dx^2}a(x) - \frac{d}{dx}[um(x)a(x)], \tag{A 11}$$

where $u$ is an appropriate constant. Normalization and boundary conditions are

$$M = \int_0^R dx\, m(x), \tag{A 12}$$

$$A = \int_0^R dx\, a(x) \tag{A 13}$$

and

$$\frac{d}{dx}a(x)\bigg|_{x=0} = 0 = \frac{d}{dx}a(x)\bigg|_{x=R}. \tag{A 14}$$

From equations (A 10) and (A 12), it follows that

$$m(x) = m_0\, e^{-(k_c/v)x}, \tag{A 15}$$

where

$$m_0 = \frac{k_c}{v}(1 - e^{-(k_c/v)R})^{-1}M.$$

From equation (A 11), instead, it follows that

$$D\frac{d}{dx}a(x) - um(x)a(x) = \text{const.}, \tag{A 16}$$

i.e. a first-order linear differential equation, the solution of which is

$$a(x) = \exp\left[-\frac{u}{D}\frac{v}{k_c}m_0\, e^{-(k_c/v)x}\right]\left[c_1 - c_2\frac{u}{D}Ei\left(\frac{u}{D}\frac{v}{k_c}m_0\, e^{-(k_c/v)x}\right)\right], \tag{A 17}$$

where $c_1$ and $c_2$ are integration constants and

$$Ei(z) = \int_{-\infty}^z dy\, \frac{e^y}{y} \tag{A 18}$$

is the exponential integral. Reflecting boundary conditions at $x = 0$ and $x = R$ imply

$$c_2 = 0, \tag{A 19}$$

while, from the normalization condition,

$$A = \int_0^R dx\, a(x) = c_1\frac{v}{k_c}\left[Ei\left(-\frac{u}{D}\frac{v}{k_c}m_0\right) - Ei\left(-\frac{u}{D}\frac{v}{k_c}m_0\, e^{-(k_c/v)R}\right)\right], \tag{A 20}$$

we obtain

$$c_1 = \frac{A}{(v/k_c)[Ei(-(u/D)(v/k_c)m_0) - Ei(-(u/D)(v/k_c)m_0\, e^{-(k_c/v)R})]}. \tag{A 21}$$

The distribution of the actin filaments is then

$$a(x) = \frac{A}{v/k_c[Ei(-(u/D)(v/k_c)m_0) - Ei(-(u/D)(v/k_c)m_0\, e^{-(k_c/v)R})]}\exp\left[-\frac{u}{D}\frac{v}{k_c}m_0\, e^{-(k_c/v)x}\right]. \tag{A 22}$$

Figure 7 shows that in one-dimension, the approximation of considering the transport of actin as the result of a velocity field that drives the filaments towards the surface of the cell is reasonable, suggesting that the discrepancy observed in the three-dimensional case emerges from the geometry of the system. Unfortunately, at present, we lack a mechanistic description of the system that would enable us to derive a suitable value of the velocity $u$ from the other model parameters, which would be a useful avenue of further research.

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
