## [Reviewer comments · Royal Society Open Science]

Review History

Decision letter (RSOS-201730.R0)

Dear Dr Saltini:

It is a pleasure to accept your manuscript entitled "Microtubule-based actin transport and localization in a spherical cell" in its current form for publication in Royal Society Open Science.

Please ensure that you send to the editorial office an editable version of your accepted manuscript, and individual files for each figure and table included in your manuscript. You can send these in a zip folder if more convenient. Failure to provide these files may delay the processing of your proof.

You can expect to receive a proof of your article in the near future. Please contact the editorial office (opencscience_proofs@royalsociety.org) and the production office (opencscience@royalsociety.org) to let us know if you are likely to be away from e-mail contact -- if you are going to be away, please nominate a co-author (if available) to manage the proofing process, and ensure they are copied into your email to the journal.

on behalf of Dr Pietro Cicuta (Subject Editor).

Editor Dr Pietro Cicuta Comments to Author:

Associate Editor: 1

Comments to the Author:

Authors have responded to the two reviewers from Roy.Soc.Interface.
